# Scaling Up Probabilistic Circuits by Latent Variable Distillation

**Anji Liu,**[*] **Honghua Zhang**[*] **& Guy Van den Broeck**
Department of Computer Science
University of California, Los Angeles
{liuanji,hzhang19,guyvdb}@cs.ucla.edu

## Abstract

Probabilistic Circuits (PCs) are a unified framework for tractable probabilistic models that support efficient computation of various probabilistic queries (e.g., marginal probabilities). One key challenge is to scale PCs to model large and high-dimensional real-world datasets: we observe that as the number of parameters in PCs increases, their performance immediately plateaus. This phenomenon suggests that the existing optimizers fail to exploit the full expressive power of large PCs. We propose to overcome such bottleneck by **latent variable distillation**: we leverage the less tractable but more expressive deep generative models to provide extra supervision over the latent variables of PCs. Specifically, we extract information from Transformer-based generative models to assign values to latent variables of PCs, providing guidance to PC optimizers. Experiments on both image and language modeling benchmarks (e.g., ImageNet and WikiText-2) show that latent variable distillation substantially boosts the performance of large PCs compared to their counterparts without latent variable distillation. In particular, on the image modeling benchmarks, PCs achieve competitive performance against some of the widely-used deep generative models, including variational autoencoders and flow-based models, opening up new avenues for *tractable* generative modeling. Our code can be found at https://github.com/UCLA-StarAI/LVD.

## 1 Introduction

The development of tractable probabilistic models (TPMs) is an important task in machine learning: they allow various tractable probabilistic inference (e.g., computing marginal probabilities), enabling a wide range of down-stream applications such as lossless compression (Liu et al., 2022) and constrained/conditional generation (Peharz et al., 2020a). Probabilistic circuit (PC) (Choi et al., 2020) is a unified framework for a wide range of families of TPMs, examples include bounded tree-width graphical models (Meila & Jordan, 2000), And-

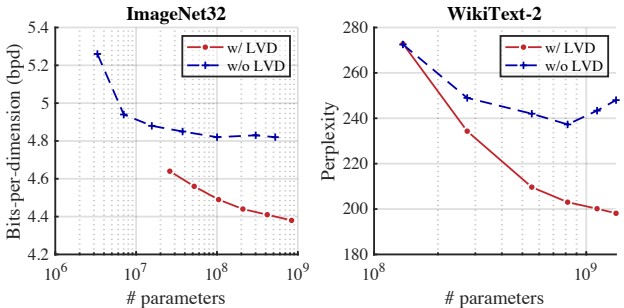

Figure 1: Latent variable (LV) distillation significantly boosts PC performance on challenging image (ImageNet32) and language (WikiText-2) modeling datasets. Lower is better.

Or search spaces (Marinescu & Dechter, 2005), hidden Markov models (Rabiner & Juang, 1986), Probabilistic Sentential Decision Diagrams (Kisa et al., 2014) and sum-product networks (Poon & Domingos, 2011). Yet, despite the tractability of PCs, scaling them up for generative modeling on large and high-dimensional vision/language dataset has been a key challenge.

By leveraging the computation power of modern GPUs, recently developed PC learning frameworks (Peharz et al., 2020a; Molina et al., 2019; Dang et al., 2021) have made it possible to train PCs with

---

[*]Authors contributed equally.

over 100M parameters (e.g., Correia et al. (2022)). Yet these computational breakthroughs are not leading to the expected large-scale learning breakthroughs: as we scale up PCs, their performance immediately plateaus (dashed curves in Fig. 1), even though their actual expressive power should increase monotonically with respect to the number of parameters. Such a phenomenon suggests that the existing optimizers fail to utilize the expressive power provided by large PCs. PCs can be viewed as latent variable models with a deep hierarchy of latent variables. As we scale them up, size of their latent space increases significantly, rendering the landscale of the marginal likelihood over observed variables highly complex. We propose to ease this optimization bottleneck by **latent variable distillation** (LVD): we provide extra supervision to PC optimizers by leveraging less-tractable yet more expressive deep generative models to induce semantics-aware assignments to the latent variables of PCs, in addition to the observed variables.

The LVD pipeline consists of two major components: (i) inducing assignments to a subset of (or all) latent variables in a PC by information obtained from deep generative models and (ii) estimating PC parameters given the latent variable assignments. For (i), we focus on a clustering-based approach throughout this paper: we cluster training examples based on their neural embeddings and assign the same values to latent variables for examples in the same cluster; yet, we note that there is no constraint on how we should assign values to latent variables and the methodology may be engineered depending on the nature of the dataset and the architecture of PC and deep generative model. For (ii), to leverage the supervision provided by the latent variable assignments obtained in (i), instead of directly optimizing the maximum-likelihood estimation objective for PC training, we estimate PC parameters by optimizing the its lower-bound shown on the right-hand side:

$$\sum\nolimits_{i=1}^{N} \log p(\boldsymbol{x}^{(i)}) := \sum\nolimits_{i=1}^{N} \log \sum\nolimits_{\boldsymbol{z}} p(\boldsymbol{x}^{(i)}, \boldsymbol{z}) \geq \sum\nolimits_{i=1}^{N} \log p(\boldsymbol{x}^{(i)}, \boldsymbol{z}^{(i)}), \qquad (1)$$

where $\{\boldsymbol{x}^{(i)}\}_{i=1}^{N}$ is the training set and $\boldsymbol{z}^{(i)}$ is the induced assignments to the latent variables for $\boldsymbol{x}^{(i)}$. After LVD, we continue to finetune PC on the training examples to optimize the actual MLE objective, i.e., $\sum_{i} \log p(\boldsymbol{x}^{(i)})$.

As shown in Figure 1, with LVD, PCs successfully escape the plateau: their performance improves progressively as the number of parameters increases. Throughout the paper, we highlight two key advantages of LVD: first, it makes much better use of the extra capacity provided by large PCs; second, by leveraging the supervision from distilled LV assignments, we can significantly speed up the training pipeline, opening up possibilities to further scale up PCs.

We start by presenting a simple example where we apply LVD on hidden Markov models to improve their performance on language modeling benchmarks (Sec. 2). Then we introduce the basics for PCs (Sec. 3.1) and present the general framework of LVD for PCs (Sec. 3.2). The general framework is then elaborated in further details, focusing on techniques to speed up the training pipeline (Sec. 4). In Section 5, we demonstrate how this general algorithm specializes to train patch-based PCs for image modeling. Empirical results show that LVD outperforms SoTA TPM baselines by a large margin on challenging image modeling tasks. Besides, PCs with LVD also achieve competitive results against various widely-used deep generative models, including flow-based models (Kingma & Dhariwal, 2018; Dinh et al., 2016) and variational autoencoders (Maaløe et al., 2019) (Sec. 6).

## 2   LATENT VARIABLE DISTILLATION FOR HIDDEN MARKOV MODEL

In this section, we consider the task of language modeling by hidden Markov models (HMM) as an illustrating example for LVD. In particular, we demonstrate how we can use the BERT model (Devlin et al., 2019) to induce semantics-aware assignments to the latent variables of HMMs. Experiments on the WikiText-2 (Merity et al., 2016) dataset show that our approach effectively boosts the performance of HMMs compared to their counterpart trained with only random initialization.

**Dataset & Model.** The WikiText-2 dataset consists of roughly 2 million tokens extracted from Wikipedia, with a vocabulary size of 33278. Following prior works on autoregressive language modeling (Radford et al., 2019), we fix the size of the *context window* to be 32: that is, the HMM model will only be trained on subsequences of length 32 and whenever predicting the next token, the model is only conditioned on the previous 31 tokens. In particular, we adopt a *non-homogeneous* HMM model, that is, its transition and emission probabilities at each position share no parameters;

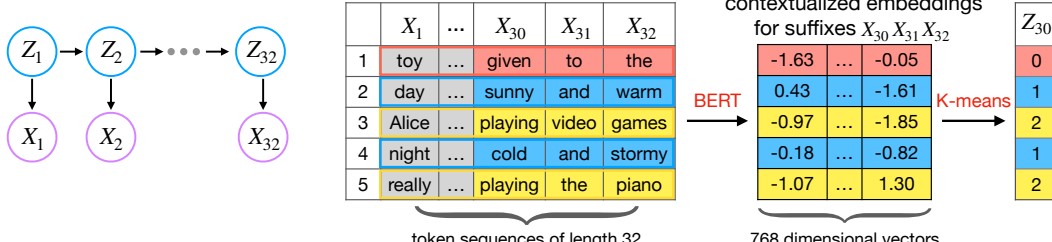

(a) Graphical model representation of an HMM modeling token sequences of length 32. $X_i$ are the observed variables and $Z_i$ are the latent variables.

(b) Pipeline for inferring values for one latent variable $Z_{30}$. We feed token sequences to the BERT model to obtain contextualized embeddings for their suffixes $X_{30}X_{31}X_{32}$; then we cluster all suffix embeddings into $h$ clusters; here $h = 3$ is the number of hidden states and the value for $Z_{30}$ is set to the cluster id. We repeat this procedure *independently* to infer values for all $Z_i$s.

Figure 2: Latent variable distillation pipeline for hidden Markov models.

Figure 2a) shows its representation as a graphical model, where $X_i$s are the observed variables and $Z_i$s are the latent variables. To facilitate training and evaluation, we pre-process the tokens from WikiText-2 by concatenating them into one giant token sequence and collect all subsequences of length 32 to construct the train, validation and test sets, respectively.

**Latent Variable Distillation.** Let $\mathcal{D} = \{\boldsymbol{x}^{(i)}\}_i$ be the training set; Figure 2 shows an example on how to induce, for each training example $\boldsymbol{x}^{(i)}$, its corresponding assignment to the latent variable $Z_{30}$. We first feed all training examples to the BERT model to compute the contextualized embeddings for their suffixes $X_{30}X_{31}X_{32}$. We cluster all suffix embeddings into $h$ clusters by the K-means algorithm (Lloyd, 1982), where $h$ is the number of hidden states; then, we set $Z_{30}$ to be the cluster id of their corresponding suffixes, that is, suffixes in the same cluster get the same latent variable value: the intuition is that if the BERT embeddings of some suffixes are close to each other then the suffixes should be relatively similar, suggesting that they should be "generated" by the same hidden state. We repeat this procedure for 32 times to infer the values for all $Z_i$s. Now we obtain an "augmented" training set $\mathcal{D}_{\mathrm{aug}} = \{(\boldsymbol{x}^{(i)}, \boldsymbol{z}^{(i)})\}_i$, where $\boldsymbol{z}^{(i)}$ are the corresponding assignments to the latent variables $\mathbf{Z}$; then, as suggested by Equation 1, we maximize the lower-bound $\sum_i \log p(\boldsymbol{x}^{(i)}, \boldsymbol{z}^{(i)})$ for the true MLE objective $\sum_i \log p(\boldsymbol{x}^{(i)})$. The parameters of the HMM that maximize $\sum_i \log p(\boldsymbol{x}^{(i)}, \boldsymbol{z}^{(i)})$, denoted by $\theta^*$, can be solved in closed-form. Finally, using $\theta^*$ as a starting point, we finetune the HMM model via EM to maximize the true MLE objective $\sum_i \log p(\boldsymbol{x}^{(i)})$.

**Experiments.** We apply LVD to HMMs with a varying number of hidden states $h = 128, 256, 512, 750, 1024$ and $1250$; for comparison, we also train HMMs with random initialization. Please refer to Appx. C for details about training. The plot on the right of Figure 1 shows the test perplexity of HMMs (w/ and w/o LVD) on WikiText-2: as the number of parameters in HMM increases, the performance of the HMMs trained with random parameter initialization immediately plateaus, while the performance of the HMMs trained with LVD progressively improves, suggesting that LVD effectively exploits the express power of the larger models.

## 3 LATENT VARIABLE DISTILLATION FOR PROBABILISTIC CIRCUITS

The previous section uses HMM as a specific TPM to elaborate key steps in LVD. In order to generalize LVD to broader TPMs, this section introduces Probabilistic Circuit (PC), which is a unifying framework for a large collection of tractable probabilistic models.

### 3.1 PROBABILISTIC CIRCUITS: A GENERAL TPM FRAMEWORK

PCs (Choi et al., 2020) are an umbrella term for a large variety of TPMs. Their syntax and semantics are defined as follows.

**Definition 1** (Probabilistic Circuits). A PC $p(\mathbf{X})$ that encodes a distribution over variables $\mathbf{X}$ is defined by a parameterized directed acyclic computation graph (DAG) with a single root node $n_r$.

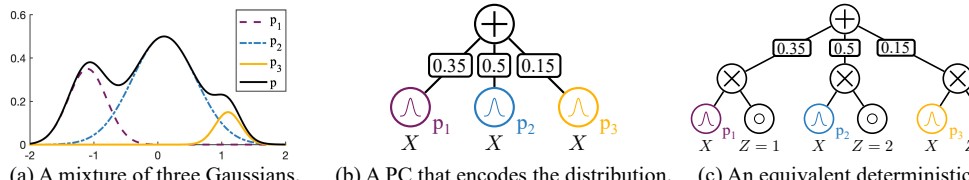

(a) A mixture of three Gaussians.   (b) A PC that encodes the distribution.   (c) An equivalent deterministic PC.

Figure 3: A mixture-of-Gaussian distribution (a) and two PCs (b-c) that encode the distribution.

Every node in the DAG corresponds to a computational unit. Specifically, every leaf node is defined by an *input unit* and every inner node $n$ represents either a *sum* or *product unit* that receives inputs from its children, termed $\mathsf{in}(n)$. Each PC unit $n$ encodes a distribution $p_n$:

$$p_n(\boldsymbol{x}) := \begin{cases} f_n(\boldsymbol{x}) & \text{if } n \text{ is an input unit,} \\ \sum_{c \in \mathsf{in}(n)} \theta_{c|n} \cdot p_c(\boldsymbol{x}) & \text{if } n \text{ is a sum unit,} \\ \prod_{c \in \mathsf{in}(n)} p_c(\boldsymbol{x}) & \text{if } n \text{ is a product unit,} \end{cases} \tag{2}$$

where $f_n(\boldsymbol{x})$ is a univariate probability distribution (e.g., Gaussian, Categorical) defined on a variable in $\mathbf{X}$ and $\theta_{c|n}$ is the parameter corresponds to edge $(n, c)$. For every sum unit $n$, we assume all its edge parameters $\{\theta_{c|n}\}_{c \in \mathsf{in}(n)}$ are non-negative and sum up to one. Intuitively, a product unit encodes a factorized distribution over its inputs, and a sum unit models a weighted mixture of its children's distributions. A PC represents the distribution encoded by its root unit $n_r$. We further assume w.l.o.g. that PCs alternate between sum and product layers before reaching an input layer.

A key property that separates PCs from many other generative models is their *tractability*, i.e., the ability to answer various queries exactly while efficiently. Such queries include common ones like marginals and conditional probabilities as well as task-specific ones such as structured prediction (Shao et al., 2022) and variational inference (Shih & Ermon, 2020). The tractability of PCs is governed by structural constraints on their DAG structure. For example, *smoothness* and *decomposability* together guarantee linear time (w.r.t. size of the PC) computation of arbitrary marginal probabilities.

**Definition 2** (Smoothness and Decomposability). Define the (variable) scope $\phi(n)$ of a PC unit $n$ as the set of variables defined by all its descendent input units. A PC is smooth if for every sum unit $n$, all its children have the same scope: $\forall c_1, c_2 \in \mathsf{in}(n), \phi(c_1) = \phi(c_2)$. A PC is decomposable if the children of every product unit $n$ have disjoint scopes: $\forall c_1, c_2 \in \mathsf{in}(n)(c_1 \neq c_2), \phi(c_1) \cap \phi(c_2) = \varnothing$.

## 3.2 MATERIALIZING AND DISTILLING LATENT VARIABLES IN PROBABILISTIC CIRCUITS

PCs can be viewed as latent variable models with discrete latent spaces (Peharz et al., 2016). Specifically, since a sum unit in a PC can be viewed as a mixture over its input distributions, it can also be interpreted as a simple latent variable model $\sum_z p(\boldsymbol{x}|z)p(z)$, where $z$ decides which input to choose from and the summation enumerates over all inputs. Figure 3 shows such an example, where the sum unit in Figure 3 (b) represents the mixture over Gaussians in Figure 3 (a).

In general, the latent space for large PCs is hierarchical and deeply nested; as we scale them up, we are in effect scaling up the size/complexity of their latent spaces, making it difficult for optimizers to find good local optima. To overcome such bottleneck, we generalize the idea presented in Section 2 and propose *latent variable distillation* (LVD). The key intuition for LVD is to provide extra supervision on the latent variables of PCs by leveraging existing deep generative models: given a PC $p(\mathbf{X})$; we view it as a latent variable model $\sum_z p(\mathbf{X}, \mathbf{Z} = \boldsymbol{z})$ over some set of latents $\mathbf{Z}$ and assume that for each training example $\boldsymbol{x}^{(i)}$, a deep generative model can always induce some semantics-aware assignment $\mathbf{Z} = \boldsymbol{z}^{(i)}$; then, instead of directly optimizing the MLE objective $\sum_i \log p(\boldsymbol{x}^{(i)})$, we can optimize its lower-bound $\sum_i \log p(\boldsymbol{x}^{(i)}, \boldsymbol{z}^{(i)})$, thus incorporating the guidance provided by the deep generative model. The LVD pipeline consists of three major steps, elaborated in the following:

**Step 1: Materializing Latent Variables.**   The first step of LVD is to *materialize* some/all latent variables in PCs. By materializing latent variables, we can obtain a new PC representing the joint distribution $\Pr(\mathbf{X}, \mathbf{Z})$, where the latent variables $\mathbf{Z}$ are explicit and its marginal distribution $\Pr(\mathbf{X})$ corresponds to the original PC. Although we can assign every sum unit in a PC an unique LV, the semantics of such materialized LVs depend heavily on PC structure and parameters,

---

**Algorithm 1** Materializing a LV in a PC

---

1: **Input:** A PC $p(\mathbf{X})$ and a variable scope $\mathbf{W}$ for some sum unit in $p(\mathbf{X})$
2: **Output:** An augmented PC $p$ defined over $\{\mathbf{X}, Z\}$, where $Z$ is the materialized LV corresponding to $\mathbf{W}$
3: $S_{\mathbf{W}} \leftarrow \{n : n \in p \text{ s.t. } n \text{ is a product unit and } \phi(n) = \mathbf{W}\}$  ▷ Created as an ordered set
4: **for** $j = 1$ **to** $|S_{\mathbf{W}}|$ **do**
5: $\quad$ Let $n_j$ be the $j$th unit in $S_{\mathbf{W}}$
6: $\quad$ Add an input unit $c$ over $Z_i$ with distribution $p_c(z_i) = \begin{cases} 1 & z_i = j, \\ 0 & \text{otherwise} \end{cases}$ as a new child of $n_j$

---

which makes it extremely hard to obtain supervision. Instead, we choose to materialize LVs based on subsets of observed variables defined by a PC. That is, each materialized LV corresponds to all PC units with a particular variable scope (cf. Def. 2). For example, we can materialize the latent variable $Z$, which corresponds to the scope $\phi(n_i)$ ($\forall i \in [3]$), to construct the PC

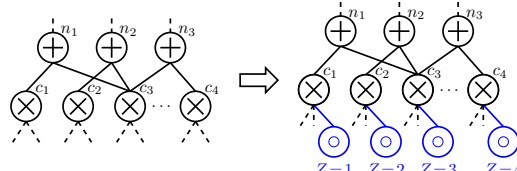

Figure 4: Materializing LVs in a PC.

in Figure 3(c) that explicitly represents $p(X, Z)$, whose marginal distribution $p(X)$ corresponds to the PC in Figure 3 (b). Algorithm 1 (Peharz et al., 2016) provides one general way to materialize latent variables in PCs, where Figure 4 shows an example where the four product units $c_1, \ldots, c_4$ are augmented with input units $Z = 1, \ldots, Z = 4$, respectively.

Continuing with our example in Figure 3, note that after materialization, the sum unit representing $p(X, Z)$ in Figure 3(c) is no longer a latent variable distribution: each assignment to $X, Z$ *uniquely* determines the input distribution to choose, where the other inputs give zero probability under this assignment; we say that this sum unit is *deterministic* (Darwiche, 2003).

**Definition 3** (Determinism). Define $\mathsf{supp}(n)$ as the set of complete assignments $\boldsymbol{x} \in \mathsf{val}(\mathbf{X})$ such that $p_n(\boldsymbol{x}) > 0$. A sum unit $n$ is deterministic if its children have disjoint supports: $\forall c_1, c_2 \in \mathsf{in}(n)(c_1 \neq c_2), \mathsf{supp}(c_1) \cap \mathsf{supp}(c_2) = \varnothing$.

Determinism characterizes whether a sum unit introduces latent variables: by materializing some sum units with the scope, we enforce them to become deterministic. Intuitively, more deterministic sum units in PCs implies smaller latent spaces, which implies easier optimization; in fact, if all sum units in a PC are deterministic then the MLE solution can be computed in closed-form (Kisa et al., 2014). By materializing more latent variables, we make PCs "more deterministic", pushing the optimization procedure towards a closed-form estimation.

**Step 2: Inducing Latent Variable Assignments.** Latent variable materialization itself cannot provide any extra supervision to the PC training pipeline; in addition, we also need to leverage some existing deep generative models to induce semantics-aware assignments for the materialized latent variables. Though there is no general guideline on how the assignments should be induced, we focus on a clustering-based approach throughout this paper. Recall from Section 2, where we cluster the suffix embeddings generated by the BERT model and for each training example, we assign the latents the cluster id that its suffixes belong to. Similarly, for image modeling, in Section 5, we will show how to induce latent variable assignments by clustering the embeddings for patches of images. The main take-away is that the method for inducing latent variable assignments should be engineered depending on the nature of the dataset and the architecture of PC and deep generative model.

**Step 3: PC Parameter Learning.** Given a PC $p(\mathbf{X}; \theta)$ with parameters $\theta$ and a training set $\mathcal{D} = \{\boldsymbol{x}^{(i)}\}$; in Step 1, by materializing some set of latent variables $\mathbf{Z}$, we obtain an augmented PC $p_{\text{aug}}(\mathbf{X}, \mathbf{Z}; \theta)$ whose marginal distribution on $\mathbf{X}$ corresponds to $p(\mathbf{X}; \theta)$; in Step 2, by leveraging some deep generative model $\mathcal{G}$, we obtain an augmented training set $\mathcal{D}_{\text{aug}} = \{(\boldsymbol{x}^{(i)}, \boldsymbol{z}^{(i)})\}$. Note that since $p_{\text{aug}}$ and $p$ share the same parameter space, we can optimize $\sum_{i=1}^{N} \log p_{\text{aug}}(\boldsymbol{x}^{(i)}, \boldsymbol{z}^{(i)}; \theta)$ as a lower-bound for $\sum_{i=1}^{N} \log p(\boldsymbol{x}^{(i)}; \theta)$:

$$\sum_{i=1}^{N} \log p(\boldsymbol{x}^{(i)}; \theta) = \sum_{i=1}^{N} \log \sum_{\boldsymbol{z}} p_{\text{aug}}(\boldsymbol{x}^{(i)}, \boldsymbol{z}; \theta) \geq \sum_{i=1}^{N} \log p_{\text{aug}}(\boldsymbol{x}^{(i)}, \boldsymbol{z}^{(i)}; \theta);$$

we denote the parameters for $p_{\text{aug}}$ after optimization by $\theta^*$. Finally, we initialize $p$ with $\theta^*$ and optimize the true MLE objective with respect to the original dataset $\mathcal{D}$, $\sum_{i=1}^{N} \log p(\boldsymbol{x}^{(i)}; \theta)$.

**Summary.** Here we summarize the general pipeline for latent variable distillation. Assume that we are given: a PC $p(\mathbf{X}; \theta)$ over observed variables $\mathbf{X}$ with parameter $\theta$, a training set $\mathcal{D} = \{\boldsymbol{x}^{(i)}\}$ and a deep generative model $\mathcal{G}$:

1. Construct a PC $p_{\text{aug}}(\mathbf{X}, \mathbf{Z}; \theta)$ by materializing a subset of latent variables $\mathbf{Z}$ in $p(\mathbf{X}; \theta)$; note that $p$ and $p_{\text{aug}}$ share the same parameter space.

2. Use $\mathcal{G}$ to induce semantics-aware latent variable assignments $\boldsymbol{z}^{(i)}$ for each training example $\boldsymbol{x}^{(i)}$; denote the augmented dataset as $\mathcal{D}_{\text{aug}} = \{\boldsymbol{x}^{(i)}, \boldsymbol{z}^{(i)}\}$.

3. Optimize the log-likelihood of $p_{\text{aug}}$ w/ respect to $\mathcal{D}_{\text{aug}}$, i.e., $\sum_i \log p_{\text{aug}}(\boldsymbol{x}^{(i)}, \boldsymbol{z}^{(i)}; \theta)$; denote the parameters for $p_{\text{aug}}$ after optimization as $\theta^*$.

4. Initialize $p(\mathbf{X}, \theta)$ with $\theta^*$ and then optimize the log-likelihood of $p$ w/ respect to the original dataset $\mathcal{D}$, i.e., $\sum_i \log p(\boldsymbol{x}^{(i)}; \theta)$.

## 4 EFFICIENT PARAMETER LEARNING

Another major obstacle for scaling up PCs is training efficiency. Specifically, despite recently developed packages (Dang et al., 2021; Molina et al., 2019) and training pipelines (Peharz et al., 2020a) that leverage the computation power of modern GPUs, training large PCs is still extremely time-consuming. For example, in our experiments, training a PC with ∼500M parameters on CIFAR (using existing optimizers) would take around one GPU day to converge. With the efficient parameter learning algorithm detailed in the following, training such a PC takes around 10 GPU hours.

The most computationally expensive part in LVD is to optimize the MLE lower bound (Eq. 1) with regard to the observed data and inferred LVs, which requires feeding all training samples through the whole PC. By exploiting the additional conditional independence assumptions introduced by the materialized LVs, we show that the computation cost of this optimization process can be significantly reduced. To gain some intuition, consider applying LVD to the PC in Figure 3(c) with materialized LV $Z$. For a sample $x$ whose latent assignment $z$ is 1, since the Gaussian distributions $p_2$ and $p_3$ are independent with this sample, we only need to feed it to the input unit corresponds to $p_1$ in order to estimate its parameters. To formalize this efficient LVD algorithm, we start by introducing the conditional independence assumptions provided by the materialized LVs.

**Lemma 1.** *For a PC $p(\mathbf{X})$, denote $\mathbf{W}$ as the scope of some units in $p$. Assume the variable scope of every PC unit is either a subset of $\mathbf{W}$ or disjoint with $\mathbf{W}$. Let $Z$ be the LV corresponds to $\mathbf{W}$ created by Algorithm 1. Then variables $\mathbf{W}$ are conditional independent of $\mathbf{X} \backslash \mathbf{W}$ given $Z$.*

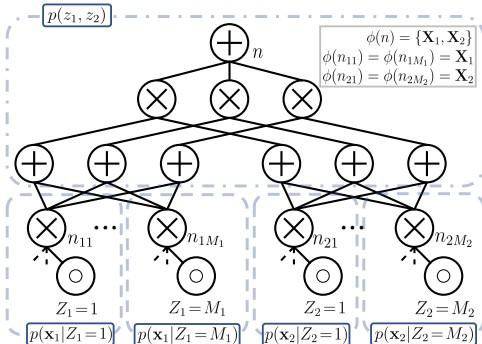

Proof of the above lemma is provided in Appx. A.1. Take Figure 4 as an example. Define the scope of $\{n_i\}_{i=1}^3$ and $\{c_i\}_{i=1}^4$ as $\mathbf{W}$ and the corresponding LV as $Z$; denote the scope of the full PC as $\mathbf{X}$. Lemma 1 implies that variables $\mathbf{W}$ and $\mathbf{X} \backslash \mathbf{W}$ are conditional independent given $Z$.

Figure 5: Distribution decomposition of an example PC with materialized LVs $Z_1, Z_2$.

We consider a simple yet effective strategy for materializing LVs: the set of observed variables $\mathbf{X}$ is partitioned into $k$ disjoint subsets $\{\mathbf{X}_i\}_{i=1}^k$; then for each $\mathbf{X}_i$, we use Algorithm 1 to construct a corresponding LV, termed $Z_i$. As a direct corollary of Lemma 1, the joint probability over $\mathbf{X}$ and $\mathbf{Z}$ can be decomposed as follows: $p(\boldsymbol{x}, \boldsymbol{z}) = p(\boldsymbol{z}) \prod_{i=1}^k p(\boldsymbol{x}_i | z_i)$.

The key to speed up LVD is the observation that the MLE lower bound objective (Eq. 1) can be factored into independent components following the decomposition of $p(\boldsymbol{x}, \boldsymbol{z})$:

$$\text{LL}(p, \mathcal{D}_{\text{aug}}) := \sum_{l=1}^N \log p(\boldsymbol{x}^{(l)}, \boldsymbol{z}^{(l)}) = \sum_{l=1}^N \sum_{i=1}^k \log p(\boldsymbol{x}_i^{(l)} | z_i^{(l)}) + \sum_{l=1}^N \log p(\boldsymbol{z}^{(l)}), \quad (3)$$

(a) Illustration of the Masked Autoencoder model.   (b) Example of image patches belonging to the same cluster.

Figure 6: Extracting LVs for image data. The MAE model (a) is used to extract categorical LVs $\{Z_i\}_{i=1}^k$ that correspond to image patches $\{\mathbf{X}_i\}_{i=1}^k$, respectively. (b) provides example patches from the training set that belong to four randomly chosen clusters of the LV $Z_1$.

where $\mathcal{D}_{\text{aug}} := \{(\boldsymbol{x}^{(l)}, \boldsymbol{z}^{(l)})\}_{l=1}^N$ is the training set augmented with LV assignments. According to Equation (3), optimize $\text{LL}(p, \mathcal{D}_{\text{aug}})$ is equivalent to performing MLE on the factorized distributions separately. Specifically, we decompose the optimization process into the following independent steps: (i) for each cluster $i$ and category $j$, optimizing PC parameters w.r.t. the distribution $p(\mathbf{X}_i | Z_i = j)$ using the subset of training samples whose LV $z_i$ is assigned to category $j$, and (ii) optimizing the sub-PC corresponds to $p(\boldsymbol{z})$ using the set of all LV assignments. Consider the example PC shown in Figure 5. The subset of PC surrounded by every blue box encodes the distribution labeled on its edge. To maximize $\text{LL}(p, \mathcal{D}_{\text{aug}})$, we can separately train the sub-PCs correspond to the decomposed distributions, respectively. Compared to feeding training samples to the whole PC, the above procedure trains every latent-conditioned distribution $p(\mathbf{X}_i | Z_i = j)$ using only samples that have the corresponding LV assignment (i.e., $z_i = j$), which significantly reduces computation cost.

Recall from Section 3.2 that in the LVD pipeline, after training the PC parameters by maximizing $\text{LL}(p, \mathcal{D}_{\text{aug}})$, we still need to finetune the model on the original dataset $\mathcal{D}$. However, this finetuning step often suffers from slow convergence speed, which significantly slows down the learning process. To mitigate this problem, we add an additional *latent distribution training* step where we only finetune parameters correspond to $p(\mathbf{Z})$. In this way, we only need to propagate training samples through the sub-PCs correspond to the latent-conditioned distributions once. After this step converges, we move on to finetune the whole model, which then takes much fewer epochs to converge.

## 5 EXTRACTING LATENT VARIABLES FOR IMAGE MODELING

This section discusses how to induce assignments to LVs using expressive generative models. While the answer is specific to individual data types, we proposes preliminary answers of the question in the context of image data. We highlight that there are many possible LV selection strategies and target generative model; the following method is only an example that shows the effectiveness of LVD.

Motivated by recent advances of image-based deep generative models (Dosovitskiy et al., 2020; Liu et al., 2021), we model images by two levels of hierarchy — the low-level models independently encode distribution of every image patch, and the top-level model represents the correlation between different patches. Formally, we define $\mathbf{X}_i$ as the variables in the $i$th $M \times M$ patch of an $H \times W$ image (w.l.o.g. assume $H$ and $W$ are both divisible by $M$). Therefore, the image $\mathbf{X}$ is divided into $k = H \cdot W / M^2$ subsets $\{\mathbf{X}_i\}_{i=1}^k$. Every $Z_i$ is defined as the LV corresponds to patch $\mathbf{X}_i$.

Recall that our goal is to obtain the assignment of $\{Z_i\}_{i=1}^k$, each as a concise representation of $\{\mathbf{X}_i\}_{i=1}^k$, respectively. Despite various possible model choices, we choose to use Masked Autoencoders (MAEs) (He et al., 2022) as they produce good features for image patches. Specifically, as shown in Figure 6(a), MAE consists of an encoder and a decoder. During training, a randomly selected subset of patches are fed to the encoder to generate a latent representation for every patch. The features are then fed to the decoder to reconstruct the full image. The simplest way to compute latent features for every patch is to feed them into the encoder independently, and extract the corresponding features. However, we find that it is beneficial to also input other patches as context. Specifically, we first compute the latent features without context. We then compute the correlation between features of all pair of patches and construct the Maximum Spanning Tree (MST) using the pairwise correlations. Finally, to compute the feature of each patch $\mathbf{X}_i$, we additionally input patches correspond to its ancestors in the MST. Further details are given in Appx. B.2.

As shown in Section 3.2, LVs $\{Z_i\}_{i=1}^k$ are required to be categorical. To achieve this, we run the K-means algorithm on the latent features (of all training examples) and use the resultant cluster

Table 1: Density estimation performance of Tractable Probabilistic Models (TPMs) and Deep Generative Models (DGMs) on three natural image datasets. Reported numbers are test set bit-per-dimension (bpd). Bold indicates best bpd (smaller is better) among all four TPMs.

| Dataset | TPMs | | | | DGMs | | |
|---|---|---|---|---|---|---|---|
| | LVD (ours) | HCLT | EiNet | RAT-SPN | Glow | RealNVP | BIVA |
| ImageNet32 | **4.39**$_{\pm 0.01}$ | 4.82 | 5.63 | 6.90 | 4.09 | 4.28 | 3.96 |
| ImageNet64 | **4.12**$_{\pm 0.00}$ | 4.67 | 5.69 | 6.82 | 3.81 | 3.98 | - |
| CIFAR | **4.38**$_{\pm 0.02}$ | 4.61 | 5.81 | 6.95 | 3.35 | 3.49 | 3.08 |

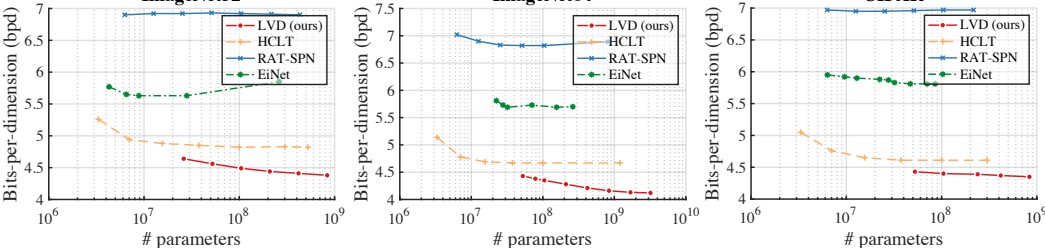

Figure 7: Generative modeling performance of four TPMs on three natural image datasets. For each method, we report the test set bits-per-dimension (y-axis) in terms of the number of parameters (x-axis) for different numbers of latent states.

indices as the LV assignments. Figure 6(b) shows some example image patches $x_1$ belonging to four latent clusters (i.e., $Z_1 = 1, \ldots, 4$). Clearly, the LVs capture semantics of different image patches.

To illustrate the effectiveness of LVD, we make minimum structural changes compared to Hidden Chow-Liu Trees (HCLTs) (Liu & Van den Broeck, 2021), a competitive PC structure. Specifically, we use the HCLT structure for all sub-PCs $\{p(x_i|Z_i=j)\}_{i,j}$ and $p(z)$. This allows us to materialize patch-based LVs while keeping the model architecture similar to HCLTs.

## 6 EXPERIMENTS

In this section, we evaluate the proposed latent variable distillation (LVD) technique on three natural image benchmarks, i.e., CIFAR (Krizhevsky et al., 2009) and two versions of down-sampled ImageNet (ImageNet32 and ImageNet64) (Deng et al., 2009). On all benchmarks, we demonstrate the effectiveness of LVD from two perspectives. First, compared to PCs trained by existing EM-based optimizers, the proposed technique offers a significant performance gain especially on large PCs. Second, PCs trained by LVD achieve competitive performance against some of the less tractable deep generative models, including variational autoencoders and flow-based models.

**Baselines** We compare the proposed method against three TPM baselines: Hidden Chow-Liu Tree (HCLT) (Liu & Van den Broeck, 2021), Einsum Network (EiNet) (Peharz et al., 2020a), and Random Sum-Product Network (RAT-SPN) (Peharz et al., 2020b). Though not exhaustive, this baseline suite embodies many of the recent advancement in tractable probabilistic modeling, and can be deemed as the existing SoTA. To evaluate the performance gap with less tractable deep generative models, we additionally compare LVD with the following flow-based and VAE models: Glow (Kingma & Dhariwal, 2018), RealNVP (Dinh et al., 2016), and BIVA (Maaløe et al., 2019).

To facilitate a fair comparison with the chosen TPM baselines, we implement both HCLT and RAT-SPN using the Julia package Juice.jl (Dang et al., 2021) and tune hyperparameters such as batch size, learning rate and its schedule. We use the original PyTorch implementation of EiNet and similarly tune their hyperparameters. For all TPMs, we train various models with number of parameters ranging from ∼1M to ∼100M, and report the number of the model with the best performance. For deep generative model baselines, we adopt the numbers reported in the respective original papers. Please refer to Appx. C for more details of the experiment setup.

**Empirical Insights** We first compare the performance of the four TPM approaches. As shown in Figure 1, for all three benchmarks, PCs trained by LVD are consistently better than the competitors

by a large margin. In particular, on ImageNet32, a $\sim$25M PC trained by LVD is better than a HCLT with $\sim$400M parameters. Next, looking at individual curves, we observe that with LVD, the test set bpd keeps decreasing as the model size increases. This indicates that LVD is able to take advantage of the extra capacity offered by large PCs. In contrast, PCs trained by EM immediately suffer from a performance bottleneck as the model size increases. Additionally, the efficient LVD learning pipeline described in Section 4 allows us to train PCs with 500M parameters in 10 hours with a single NVIDIA A5000 GPU, while existing optimizers need over 1 day to train baseline PCs with similar sizes. Please refer to Appx. D for detailed analysis of the computation efficiency of LVD.

We move on to compare the performance of LVD with the three adopted DGM baselines. As shown in Table 1, although the performance gap is relatively large on CIFAR, the performance of LVD is highly competitive on ImageNet32 and ImageNet64, with bpd gap ranging from $\sim$0.1 to $\sim$0.3. We hypothesize that the relatively large performance gap on CIFAR is caused by insufficient training samples. Specifically, for the PC structures specified in Section 5, the sub-PCs correspond to the latent-conditioned distributions $\{p(\boldsymbol{x}_i|Z_i = j)\}_{i,j}$ are constructed independently, and thus every training sample $\boldsymbol{x}_i$ can only be used to train its corresponding latent-conditioned distribution, making the model extremely data-hungry. However, we note that this is not an inherent problem of LVD. For example, by performing parameter tying of sub-PCs correspond to different image patches, we can significantly improve sample complexity of the model. This is left to future work.

## 7    RELATED WORKS

There has been various recent endeavors to improve the performance of PCs on modeling complex and high-dimensional datasets. An extensively-explored direction is to construct or learn the structure of PCs that is tailored to the target dataset. For example, Gens & Pedro (2013); Dang et al. (2022) seek to progressively improve the PC structure during the optimization process. Many other work aim to construct good PC structures given the dataset in one shot, and then move on to parameter optimization (Rahman et al., 2014; Adel et al., 2015). Model agnostic PC structures such as RAT-SPN (Peharz et al., 2020b) and extremely randomized PCs (XPCs) (Di Mauro et al., 2021) are also shown effective in various density estimation benchmarks. There are also papers that focus exclusively on scaling up a particular TPM. For example, scaling HMM (Chiu & Rush, 2020) uses techniques such as learning blocked emission probabilities to boost the performance of HMMs.

Another line of work seek to scale up PCs by learning hybrid models with neural networks (NNs). Specifically, Shao et al. (2022) leverages the expressive power of NNs to learn expressive yet tractable conditional distributions; HyperSPN (Shih et al., 2021) uses NN to regularize PC parameters, which prevents large PCs from overfitting. Such hybrid models are able to leverage the expressiveness of NNs at the cost of losing tractability on certain queries.

## 8    CONCLUSION

Scaling probabilistic circuits to large and high-dimensional real-world datasets has been a key challenge: as the number of parameters increases, their performance gain diminishes immediately. In this paper, we propose to tackle this problem by latent variable distillation: a general framework for training probabilistic circuit that provides extra supervision over their latent spaces by distilling information from existing deep generative models. The proposed framework significantly boosts the performance of large probabilistic circuits on challenging benchmarks for both image and language modeling. In particular, with latent variable distillation, a image-patch-structured probabilistic circuit achieves competitive performance against flow-based models and variational autoencoders. Despite its empirical success on scaling up probabilistic circuits, at high-level, latent variable distillation also implies a new way to organically combine probabilistic circuits and neural models, opening up new avenues for tractable generative modeling.

**Reproducibility statement**    To facilitate reproducibility, we provide detailed description of the models and training details in both the main text and the appendix. Specifically, the last paragraph in Section 5 elaborates the PC structure used for image modeling tasks; training details of both the proposed method and the baselines are provided in Section 6 and Appx. C. For baselines, we always use the official GitHub implementation if possible. For our method, we provide detailed explanation of all hyperparameters used in the experiment (Appx. C).

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

# A  Proofs

In this section, we provide detailed proofs of Lemma 1.

## A.1  Proof of Lemma 1

*Proof.* Define $\mathbf{Y} := \mathbf{X}\backslash\mathbf{W}$. To prove that variables $\mathbf{W}$ is conditional independent with $\mathbf{Y}$ given $Z$, it is sufficient to show that $\forall\mathbf{w}\in\mathsf{val}(\mathbf{W}), z\in\mathsf{val}(Z), \boldsymbol{y}\in\mathsf{val}(\mathbf{Y})$, we have $p(\mathbf{w}|z) = p(\mathbf{w}|z,\boldsymbol{y})$.

Define $S_{p,\mathbf{W}}^{\mathrm{sum}}$ and $S_{p,\mathbf{W}}^{\mathrm{prod}}$ as the set of sum and product units with scope $\mathbf{W}$, respectively. $\forall n\in S_{p,\mathbf{W}}^{\mathrm{sum}}$, since the scope of $n$ does not contain variables $\mathbf{Y}$, we can immediately conclude that $p_n(\mathbf{w}|z) = p(\mathbf{w}|z,\boldsymbol{y})$. In order to show that this equation also holds for the root PC unit, we only need to show that for each PC unit $n$ in $p$, if all its children (whose scope contains $\mathbf{W}$) satisfy this equation, then $n$ also does. The reason is that the root unit $n_r$ of $p$ must be an ancestor unit of every unit in $S_{p,\mathbf{W}}^{\mathrm{sum}}$.

We start with the case that $n$ is a product unit. Since the PC is assumed to be decomposable, only one child, denoted $m$, satisfies $\mathbf{W}\subseteq\phi(m)$. Therefore, the distribution of $n$ can be written as

$$p_n(\boldsymbol{x}) = p_m(\boldsymbol{x})\cdot\prod_{c\in\mathsf{in}(n),c\neq m}p_c(\boldsymbol{x}) \stackrel{(a)}{=} p_m(\boldsymbol{x})\cdot\prod_{c\in\mathsf{in}(n),c\neq m}p_c(\boldsymbol{y}),$$

where $(a)$ holds because $\forall c\in\mathsf{in}(n), c\neq m$, we have $\phi(c)\cap\mathbf{W} = \varnothing$. Therefore, we have $p_n(\mathbf{w}|z) = p_m(\mathbf{w}|z)$ and $p_n(\mathbf{w}|z,\boldsymbol{y}) = p_m(\mathbf{w}|z,\boldsymbol{y})$. Taking the two equations together and use the assumption from the induction step: $p_m(\mathbf{w}|z) = p_m(\mathbf{w}|z,\boldsymbol{y})$, we conclude that $p_n(\mathbf{w}|z) = p_n(\mathbf{w}|z,\boldsymbol{y})$.

Define $n_z$ as the product unit in $S_{p,\mathbf{W}}^{\mathrm{prod}}$ that is augmented with input unit $Z = z$ by Algorithm 1. Before proving the main result, we highlight that $\forall n$ whose scope contains $\mathbf{W}$, $p_n(\boldsymbol{x})$ can be written as $p_{n_z}(\mathbf{w})\cdot g_n(\boldsymbol{y})$, where $g_n(\boldsymbol{y})$ is independent with $\mathbf{w}$. This is because $\forall m\in S_{p,\mathbf{W}}^{\mathrm{prod}}$ and $m\neq n_z$, $p_m(\mathbf{w},z) = 0(\forall\mathbf{w}\in\mathsf{val}(\mathbf{W}))$.

Next, assume $n$ is a sum unit whose scope contains $\mathbf{W}$. Using the above result, we know that every child $c$ of $n$ satisfies the following: $p_c(\boldsymbol{x}) = p_{n_z}(\mathbf{w})\cdot g_c(\boldsymbol{y})$. Thus, we have

$$p_n(\boldsymbol{x}) = \sum_{c\in\mathsf{in}(n)}\theta_{c|n}\cdot p_{n_z}(\mathbf{w})\cdot g_c(\boldsymbol{y}) = p_{n_z}(\mathbf{w})\cdot\Big(\sum_{c\in\mathsf{in}(n)}\theta_{c|n}\cdot g_c(\boldsymbol{y})\Big).$$

Since $p_{n_z}(\mathbf{w}|z) = p_{n_z}(\mathbf{w}|z,\boldsymbol{y})$, we have $p_n(\mathbf{w}|z) = p_n(\mathbf{w}|z,\boldsymbol{y})$.

Taking the above two inductive cases (i.e., for sum and product units, respectively), we can conclude that for the root unit $n_r$, $p_{n_r}(\mathbf{w}|z) = p_{n_r}(\mathbf{w}|z,\boldsymbol{y})$. □

# B  Details for Latent Variable Distillation

This section provides additional details for latent variable distillation (LVD), including description of the adopted EM algorithm and details of the LV extraction step.

## B.1  Parameter Estimation

We adopt a stochastic mini-batch version of the Expectation-Maximization algorithm. Specifically, a mini-batch of samples are drown from the dataset, and the EM algorithm for PCs (Choi et al., 2021; Dang et al., 2021) is used to compute a set of new parameters $\boldsymbol{\theta}^{\mathrm{new}}$, which is updated with a learning rate $\alpha$: $\boldsymbol{\theta}_{t+1}\leftarrow\alpha\cdot\boldsymbol{\theta}^{\mathrm{new}} + (1-\alpha)\cdot\boldsymbol{\theta}_t$.

## B.2  Details of the MAE-based LV Extraction Step

We use the official code (https://github.com/facebookresearch/mae) to train MAE models on the adopted datasets (i.e., CIFAR, ImageNet32, and ImageNet64). At each training step, the percentage of masked patches is chosen uniformly from 10% to 90%. After training, the LV extraction step follows the description in Section 5.

## C  EXPERIMENT DETAILS

In this section, we describe experiment details of all four TPMs adopted in Section 2 and Section 6.

**Hardware specification**     All experiments are run on servers/workstations with the following configuration:

- 32 CPUs, 128G Mem, $4 \times$ NVIDIA A5000 GPU;

- 32 CPUs, 64G Mem, $1 \times$ NVIDIA GeForce RTX 3090 GPU;

- 64 CPUs, 128G Mem, $3 \times$ NVIDIA A100 GPU.

**HMM**     The HMM models are trained with varying hidden states $h = 128, 256, 512, 750, 1024$ and 1250, with and without LVD. All HMM models are trained with mini-batch EM (Appx. B.1) for two phases: in phase 1, the model is trained with learning rate 0.1 for 20 epochs; in phase 2, the model is trained with learning rate 0.01 for 5 epochs. Note that for HMM models with hidden states $\geq 750$, we train for 30 epochs in phase 1. The number of epochs are selected such that all model converges before training stops.

**LVD**     For every subset $\mathbf{X}_i$, the number of hidden categories, i.e., $\{M_i\}_{i=1}^k$ are set to values in $\{8, 16, 32, 64, 128, 256\}$. For the latent-conditioned distribution $\{p(\mathbf{X}_i | Z_i = j)\}_{i,j}$, we adopt HCLTs with hidden size 16, and for the latent distribution $p(\mathbf{Z})$, a HCLT with hidden size $M_i$ is adopted. When optimizing the model with the MLE lower bound, we adopt mini-batch EM (Appx. B.1) with learning rate annealed linearly from 0.1 to 0.01. In the latent distribution training step (Sec. Section 4), we anneal learning rate from 0.1 to 0.001.

**HCLT**     We use the publicly available implementation of HCLT at `https://github.com/Juice-jl/ProbabilisticCircuits.jl/blob/master/src/structures/hclts.jl`. The hidden size is chosen from $\{16, 32, 64, 128, 256, 512, 1024\}$. We anneal the EM learning rate from 0.1 to 0.01 and train for 100 epochs, and then anneal the learning rate from 0.01 to 0.001 and train for another 100 epochs.

**RAT-SPN**     We adopt the publicly available implementation at `https://github.com/Juice-jl/ProbabilisticCircuits.jl/blob/master/src/structures/rat.jl`. num_nodes_region, num_features, and num_nodes_leaf are set to the same value, which is chosen from $\{16, 32, 64, 128, 256, 512, 1024\}$. Learning rate schedule is same with HCLTs.

**EiNet**     We use the official implementation on GitHub: `https://github.com/cambridge-mlg/EinsumNetworks`. We use the PD structure provided in the codebase. We select hyperparameter delta from $\{4, 6, 8\}$ and select num_sums from $\{16, 32, 64, 128, 256\}$. Learning rate is set to 0.001.

## D  EFFICIENCY ANALYSIS

This section provides the breakdown of the runtime for each stage of the LVD algorithm for the PC that achieves 4.38 bpd on ImageNet32. The PC has 836M parameters. All experiments are done on a single NVIDIA A5000 GPU.

For this PC, training all latent conditioned distributions $\{p(\boldsymbol{x}_i | Z_i = j)\}_{i,j}$ take $\sim 8$ hours, and training the latent distribution $p(\boldsymbol{z})$ takes $\sim 0.5$ hours. Finally, the fine-tuning stage takes $\sim 1$ hour.

As shown in the above computation time breakdown, the most time-consuming part is to train the latent conditioned distributions. However, we note that this is not a fundamental problem of LVD: we are training every latent conditioned distributions *independently*, while there could be massive structure/parameter sharing among such distributions. We left this to future work.

