# OpenReview forum: "Scaling Up Probabilistic Circuits by Latent Variable Distillation"
_ICLR.cc/2023/Conference — ICLR 2023 notable top 5%_

### Official Review · Reviewer_7gAx · 2022-10-24

**Confidence:** 4
**Correctness:** 4
**Technical Novelty And Significance:** 4
**Empirical Novelty And Significance:** 4
**Recommendation:** 8

**Clarity, Quality, Novelty And Reproducibility:**

**Clarity and quality.** I find the paper very well written and of high quality. There are a number of typos that should be corrected for the next revision. I also find the paper to reiterate the same statements too many times, but this is personal taste.

**Novelty.** The method seems novel to my eyes, and the advantages are clear. However, I find _really_ surprising that there are no citations at all regarding Probabilistic Circuits. They were first introduced in a UAI tutorial, and afterwards [a technical report](https://web.cs.ucla.edu/~yjchoi/publications/ProbCirc20/) was written by the authors which can be cited.

**Reproducibility.** The experimental set-up should be reproducible.

**Strength And Weaknesses:**

**Strengths:**
- S1. This work clearly demonstrates a novel method to exploit all the capacity of PCs using auxiliary and non-tractable methods from Deep Learning. This is extremely relevant for the PC community to close the gap with existing DL approaches.
- S2. The practical speed-up obtained by their method (section 4) is quite relevant as well.
- S3. Experimental results are strong.

**Weaknesses:**
- W1. While the idea is novel, I felt a bit let down when I read that the approach need to be "engineered" depending on the dataset, architecture, and deep generative model. The two instantiations of the approach look really ad-hoc. I would've hoped to see some sort of general guidelines, or desiredata for the method to be successful.
- W2. On a similar note, I feel a deeper ablation study would help a lot. There are too many questions open:
   - How important is the deep generative architecture to obtain good results?
   - Do other PCs architectures benefit from the proposed approach? So far, it is only tried in a modified version of HCLT.
   - K-means is also prone to fall into local optima. How much do the results vary with different K-means initializations?
   - How does the algorithm perform if we condition (Fig. 5) on deeper layers?
- W3. I might have missed it, but it is not clear to me how many times each experiment was repeated. In any case, standard deviations are not reported.
- W4. No samples were reported, so one cannot assess the qualitative improvements. Are the new generated samples better as well?

**Questions:**
- Q1. Any intuition on why the order to generate the latent samples is reversed with respect to the HMM in the example of section 2?
- Q2. Could you clarify why is the upper part of Fig. 5 supposed to be $p(z_1, z_2)$. While I agree on the bottom modules, I do not see how the top part represents the marginal of $z$.
- Q3. I understand LVD is HCLT using the proposed training approach. Why is there so much difference in number of parameters between both approaches in Figure 7? Is it because of the "minimum modifications required" described in the text? Why not comparing with that modified architecture without LVD?
-  Q4. What do you mean by "semantics" in the manuscript? I really struggle to understand what "semantic-aware" means.


**Summary Of The Paper:**

Probabilistic circuits (PCs) are a unified framework that encompasses a number of tractable probabilistic models, such as arithmetic circuits and sum-product networks. While recent work has been able to scale them by means of parallelization and vectorization, their performance does not scale with the number of parameters. This rather unintuitive results suggest that the problem comes from the optimization process, which cannot fully utilize the model parameters and gets stuck in local optima. This work proposes to leverage existing deep generative models to assist the training of PCs through a novel method called latent variable distillation. In short, the PC is extended to include a set of latent variables, which are instantiated via the latent representation from deep generative models. This alone permits to better exploit the model. Additionally, said latent extension permits to divide the PCs into conditionally independent modules which can be trained simultaneously, speeding up the training process. The authors demonstrate the advantages of the proposed method in the text and image domains by leveraging Bert and MAE models.

**Summary Of The Review:**

The paper is novel and well-written. The empirical results clearly demonstrate the advantages of the proposed method with respect to existing approaches. However, I feel the paper falls short when it comes to understanding the influence of the different factors of the proposed method in the final performance.

---

> ### Author Response · Authors · 2022-11-19
> **Response to Reviewer 7gAx (part 1/2)**
>
> We thank the reviewer for their constructive feedback. Please find detailed responses below.
>
> > While the idea is novel, I felt a bit let down when I read that the approach need to be "engineered" depending on the dataset, architecture, and deep generative model.
>
> Though we only demonstrate LVD with specific design choices in our experiments (e.g., PC architectures, how assignments LVs are induced etc), we still observe that LVD works well with other design choices. For example, as will be detailed in the following ablation study part, we observe that changing the MAE model to VQ-VAE leads to similar bpd on ImageNet32. We will also show in the following ablation studies that LVD works well across different base PC structures.
>
> Although LVD can provide significant performance boost to PCs across various settings, similar to neural networks, careful data-dependent design and tuning is important to maximize the performance gain. For example, in the patch-based model described in Section 5, we leverage the common assumption of spatial invariance used in image data to define the materialized LVs.
>
> > On a similar note, I feel a deeper ablation study would help a lot
>
> We conducted detailed ablation studies to provide a better understanding of the scope of the proposed LVD approach.
>
> - Importance of the deep generative model architecture
>
> In addition to the MAE model, we apply LVD using LV assignments generated by a pretrained VQ-VAE model [1]. Specifically, the encoder of the adopted VQ-VAE transforms every 4*4 image patch directly into a discrete latent code. The discrete codes are used directly as LV assignments by LVD. On ImageNet32, with 256 categories for each latent variable (same with the best adopted MAE model), VQ-VAE + LVD achieved 4.44 bpd. This is significantly better than the performance of the best TPM/PC without LVD, and is only slightly worse than the MAE + LVD. This result shows that LVD works well across different deep generative model architectures.
>
> - LVD + other PC architectures
>
> To evaluate whether LVD is capable of scaling up other PC architectures, we replace all HCLTs used in the patch-based image model described in Section 5 with RAT-SPNs. On ImageNet32, the RAT-SPN-based structure achieved 4.99 test bpd, which is a huge boost compared to the 6.90 test bpd w/o LVD. Therefore, LVD is able to effectively train large PC with different structures.
>
> - Performance variation with different kmeans initializations
>
> In the updated Table 1, we have included standard deviations of our method over 5 runs. In each round, kmeans starts with different initializations. Results show that kmeans initializations cause little variation on the final performance.
>
> - Condition on deeper layers
>
> In the original experiments, we select patch size 4 for the MAE model. That is, every 4*4 patch is materialized as a LV $Z$. To study the influence caused by the number of materialized LVs, we perform LVD using two new MAE models with patch size 2 and 8, respectively. Results are shown in the table below.
>
> ```text
> +================+======+=====+======+
> + patch size     +  2   +  4  +  8   +
> + ImageNet32 bpd + 4.57 + 4.38+ 4.44 +
> +================+======+=====+======+
> ```
>
> We observe that with patch size 8, the performance becomes slightly worse and with patch size 2 the performance becomes much worse. However, we highlight that the inferior performance of the patch-size-2 model is not because the use of more materialized latent variables. The reason is that the sub-PCs $p(\mathbf{x}_i \vert Z_i = j)$ suffer from worse likelihoods since the receptive field is limited to 2*2. In addition, the 2*2 patch MAE is harder to train and has larger RMSE compared to its 4*4 variant. Therefore, the performance of LVD is also determined by the deep generative model and the strategy to choose LVs.

---

> > ### Author Response · Authors · 2022-11-19
> > **Response to Reviewer 7gAx (part 2/2)**
> >
> > > I might have missed it, but it is not clear to me how many times each experiment was repeated. In any case, standard deviations are not reported.
> >
> > In the updated paper, we have included standard deviation over 5 runs for our method in Table 1. The additional runs do not change the main result of the paper as the standard deviations are relatively small. Due to limits in computational resources, we are still repeating the results for the baselines, and will update them in the next version of the paper.
> >
> > > Any intuition on why the order to generate the latent samples is reversed with respect to the HMM in the example of section 2?
> >
> > By the conditional independence property of HMMs, ${\Pr}(X_{32}, \cdots X_{i} | Z_{i}) = {\Pr}(X_{32}, \cdots X_{i} | Z_{i}, X_{i - 1}, \cdots, X_{1})$; intuitively, the value of $Z_{i}$ by itself encodes the information of the prefix $X_{1} \cdots X_{i-1}$ and determines the conditional distribution of the suffix $X_{i}, \cdots X_{32}$. Hence, if the embeddings of some suffixes are put into the same group by K-means, then these suffixes should be similar and are likely to be generated by the same hidden state $Z_i = z$.
> >
> > > Could you clarify why is the upper part of Fig. 5 supposed to be $p(z_1, z_2)$. While I agree on the bottom modules, I do not see how the top part represents the marginal of $z$
> >
> > To see the equivalence between the top part of Figure 5 and $p(\mathbf{z})$, let us consider materializing the LVs $\mathbf{Z}$ in the PC. That is, each $n_{ij}$ will have a new input node with the indicator distribution $Z_i = j$ (as already done in Figure 5). Now, if we query the PC with some $\mathbf{z}$, i.e., computing $p(\mathbf{z})$, for each $n_{ij}$, all its children except the added input node will output probability 1. Therefore, $p(\mathbf{z})$ (in this case, $\mathbf{z} = \{z_1, z_2\}$) is fully specified by the top part.
> >
> > > I understand LVD is HCLT using the proposed training approach. Why is there so much difference in number of parameters between both approaches in Figure 7? Is it because of the "minimum modifications required" described in the text? Why not comparing with that modified architecture without LVD?
> >
> > Yes, the differences between the PCs that use LVD and the plain HCLTs are caused by the “minimum required modifications” described in the last paragraph in Section 4. Specifically, each sub-PC in $\{p(\mathbf{x}_i \vert Z_i = j)\}_{i,j}$ and $p(\mathbf{z})$ are initialized as an HCLT. Thus the whole model is stacked with multiple HCLTs.
> >
> > In the previous version of the paper, we did not compare against this modified version of HCLT because its structure is dependent on the LV assignments provided by the DGM (in our case, the MAE model). Specifically, as described in Section 4, the sub-PCs $\{p(\mathbf{x}_i \vert Z_i = j)\}_{i,j}$ are pretrained with different subsets of the original dataset. Since HCLT is a data-dependent structure, the structure of LVD PCs depends on the MAE model as well. Despite the structural changes, we ran additional experiments to re-initialize the LVD PCs and train them with the existing EM-based optimizer. The PC achieved 4.68 bpd, which is better than the original HCLT (4.82 bpd) but worse than the LVD PC (4.39 bpd). On one hand, this indicates that structural change brought by LVD is beneficial. On the other hand, the parameter initialization performed by LVD is still crucial to achieving high performance.
> >
> > > What do you mean by "semantics" in the manuscript? I really struggle to understand what "semantic-aware" means
> >
> > We use the word “semantics” in contrast to “surface forms”; the main intuition is that neural embeddings for images/texts can capture the meaning of image patches/text segments beyond their surface forms. In language modeling, for example, the surface form of the two sentences, (1) “Alice is very happy today” and (2) “Alice is not very happy today” are very similar but they have opposite meanings and the intuition is that neural embeddings could potentially capture such differences in semantics.
> >
> > > The method seems novel to my eyes, and the advantages are clear. However, I find really surprising that there are no citations at all regarding Probabilistic Circuits.
> >
> > We thank the reviewer for pointing out this missing reference. We have added it in the revised manuscript.
> >
> > [1] Van Den Oord, Aaron, and Oriol Vinyals. "Neural discrete representation learning." Advances in neural information processing systems 30 (2017).

---

> > > ### Comment · Reviewer_7gAx · 2022-12-11
> > > **After-rebuttal reply**
> > >
> > > Dear authors, thank you for your response and pardon my late reply.
> > >
> > > Most of my concerns were successfully addressed. Congratulations for the great paper, and I hope to see the mentioned ablation study included in the next version of the manuscript (I couldn't find it). Similarly, I got surprised by the reference in your last reply, but I checked the updated manuscript and you indeed referenced Choi et al.

---

### Official Review · Reviewer_tcWu · 2022-10-25

**Confidence:** 5
**Correctness:** 4
**Technical Novelty And Significance:** 3
**Empirical Novelty And Significance:** 3
**Recommendation:** 8

**Clarity, Quality, Novelty And Reproducibility:**

The paper is easy to read, and introduces the idea clearly. It is a novel approach that makes TPMs even more scalable.

The algorithms presented are well defined and allows for ease of reproducibility.

**Strength And Weaknesses:**

The authors present a well-written and technically sound paper.

The empirical evaluation is outstanding for TPMs, both in terms of scale and model performance.

It would be nice to have some more comments on the impact of LVD with regards to the rest of the model. How would you do imputation? How is the performance of your model when you marginalize the LVs?



**Summary Of The Paper:**

This paper presents an approach to scale tractable probabilistic models via latent variable extensions. In particular, the authors extend the naive factorizations of models such as an SPN to introduce a new latent variable that comes from the embedding of a more complex neural network.

This extended model retains the tractability properties while reducing the bits-per-dimension for both image and text datasets compared to other state-of-the-art tractable probabilistic models.

The authors present a distributed learning approach that uses the independencies induced by the conditioning to accelerate the training, scaling the capabilities of TPMs.


**Summary Of The Review:**

I find the paper very interesting and it opens the door to integrate other more complex models while retaining some of the tractability properties.

This is a significant contribution, although I'm wondering about downside of this approach or some comments on the questions raised regarding the use of the model when we don't have access or can't compute the LVs.

The bpd results are very impressive. I'm also wondering about how this model differs from the one presented by Shao 2022, in the case you would set the latent variables and the conditional part.

---

> ### Author Response · Authors · 2022-11-19
> **Response to Reviewer tcWu**
>
> We thank the reviewer for their constructive feedback. Please find detailed responses below.
>
> > This is a significant contribution, although I'm wondering about downside of this approach or some comments on the questions raised regarding the use of the model when we don't have access or can't compute the LVs.
>
> One major assumption of the LVD pipeline is that we are able to induce reasonable assignments to latent variables. In particular, the approach we adopted for language/image modeling is to assign values to LVs by clustering neural embeddings extracted from DGMs; the inferred assignments to LVs are only used to guide parameter learning in the training stage, and are not needed when we perform inference with the model.
>
> > It would be nice to have some more comments on the impact of LVD with regards to the rest of the model. How would you do imputation? How is the performance of your model when you marginalize the LVs?
>
> Once the model is learned by LVD, this is a PC that is the same as any other PCs; no DGM components are needed after the PC is trained. As discussed in the previous paragraph, LVD is a technique that guides PC parameter learning by inferring meaningful assignments to its latent variables, and we do not need to compute LV assignments from DGMs when performing inference tasks such as imputation and marginalization; in particular, the results reported in our paper are the unconditional log-likelihoods for generative modeling.
>
> > I'm also wondering about how this model differs from the one presented by Shao 2022, in the case you would set the latent variables and the conditional part
>
> LVD and the conditional SPN (CSPN) model introduced by Shao 2022 aim to solve different problems in generative modeling. Specifically, CSPN is a new class of probabilistic models that represent tractable conditional distributions p(y|x); they are tractable in the sense that inference on y is tractable conditioned on x. In contrast, LVD is an optimization technique for PCs representing p(x) that does not affect their tractability.

---

### Official Review · Reviewer_o5Cj · 2022-10-28

**Confidence:** 4
**Correctness:** 3
**Technical Novelty And Significance:** 4
**Empirical Novelty And Significance:** 4
**Recommendation:** 8

**Clarity, Quality, Novelty And Reproducibility:**

The paper is well written with illustrative examples. I found the idea to be novel and the authors have detailed their experimental setups for reproducibility.

**Strength And Weaknesses:**

Strengths:
1) a novel method to improve the performance of large scale probabilistic circuits.
2) motivating empirical evaluations on three image datasets to show the performance improvements over very large circuits.
3) clear writeup with good examples.
Weaknesses:
I didn't find major weaknesses in the technical aspects of the paper. Please see some questions and comments below.


**Summary Of The Paper:**

This paper is about improving the expressivity of large scale probabilistic circuits (PCs). Finding a good starting point for EM based learning of these large latent variable models is problematic and the authors propose one such solution to this problem.  The main idea is to obtain semantic-aware assignments (called supervision) to the latent variables from less tractable deep generative models and then perform maximum likelihood learning over the data combined with these newly assigned latent variables. The variables that receive these assignments are said to be materialized and the assignments themselves are generated by a deep generative model by clustering over the latent embeddings of the observed (sub)space(s). When all the latent variables have been assigned values, the optimization (MLE) can be performed in closed form. The result of MLE serves as a starting point for optimizing the data likelihood in the following steps. The authors propose a couple of techniques to efficiently compute the MLE parameters which include exploiting conditional independency achieved by materialized latent variables and fine tuning the latent distributions only while keeping the parameters learned over the observed space fixed. On CIFAR and ImageNet datasets, the proposed method has shown superior performance compared to other SoTA TPMs learners and were comparable to less tractable but expressive flow-based models and VAEs.


**Summary Of The Review:**

The authors have addressed an important practical issue with large scale probabilistic circuits. The expressivity of these models tend to plateau once a certain capacity is reached typically in the order of millions of parameters. With such large scale circuits of deeply nested latent variables, the optimization landscape becomes very complex and finding a local minima becomes hard. The main idea in the paper is to make latent variables observed by assigning them values and performing an optimization step that works on less number of latent variables. This will give the EM step a good starting point. I found the idea to obtain latent variable assignments using a deep generative model to be interesting. The method seems to be effective according to the empirical results presented in the paper.
Questions:
a) In the introduction it is stated that the expressive power of PCs should monotonically increase with respect to the number of parameters. I am curious if these models don't suffer from overfitting issues. Maybe the authors could comment on this aspect.
b) Did all the models have the same structure?
c) Could the method be useful for smaller scale PCs? It seems that the clustering in the latent embedding is the key reason behind the performance boost of LVDs.
d) The significant speed up in training (for all the datasets) should probably be presented in the paper since it is mentioned in the introduction.
e) Have you done any analysis on the number of LVs that were materialized and the performance of the PCs?

---

> ### Author Response · Authors · 2022-11-19
> **Response to Reviewer o5Cj**
>
> We thank the reviewer for their detailed feedback. Please find detailed responses in the following.
>
> > I am curious if these models don't suffer from overfitting issues
>
> Like other machine learning models, PCs do suffer from overfitting issues. For example, Shih et al. [1] observe that SPNs (PCs with smoothness and decomposability) suffer from overfitting. Prior work (e.g., [1,2]) have studied various regularization techniques to mitigate overfitting issues.
>
> Although overfitting does exist in large-scale PCs, the main bottleneck in training them is still underfitting — the existing EM-based optimizers are not able to achieve good training likelihoods on PCs with more than several millions of parameters. Thus, the main focus of this paper is to develop an approach to effectively train large-scale PCs. We adopted regularization techniques such as adding pseudocounts as suggested in prior work.
>
> > Did all the models have the same structure?
>
> The main difference between the LVD PCs reported in Figure 7 is their width. There are other minor differences due to the adopted HCLT structure, as explained in the following. As described in Section 5, for image modeling, we use the HCLT structure for all sub-PCs ${ p(\mathbf{x}_i | Z_i = j) }$ and $p(\mathbf{z})$.
> Note that the structure of HCLTs depends on the provided dataset. As shown in Section 4, to develop an efficient latent variable distillation (LVD) pipeline, we train every sub-PC $p(\mathbf{x}_i \vert Z_i = j)$ with the subset of training samples whose LV $z_i$ is assigned to category $j$. Therefore, since the samples used to train each sub-PC are different, the structure of the models has differences as determined by the HCLT structure.
>
> > Could the method be useful for smaller scale PCs? It seems that the clustering in the latent embedding is the key reason behind the performance boost of LVDs
>
> LVD is more effective when used to train large-scale PCs. As shown in Figure 1, there is ~45 perplexity gain for a PC (with HMM structure) with ~10^9 parameters when using LVD, while the performance gap becomes less significant for smaller PCs.
>
> However, we highlight that this is \emph{not} necessarily a weakness of LVD, since there are aspects that render the PCs unnecessarily large. For example, a well-adopted assumption in image modeling is spatial invariance. However, in the adopted PCs, no parameter tying is used to leverage this assumption. Such approaches could significantly decrease size of the PC and do not conflict with the use of LVD. Therefore, with further development of PC structures, LVD could be used more effectively in smaller PCs.
>
> > The significant speed up in training (for all the datasets) should probably be presented in the paper
>
> Efficiency analysis of LVD is provided in the 4th paragraph of the experiment section. In the updated paper, we have added a detailed analysis of the speedup in Appendix D.
>
> > Have you done any analysis on the number of LVs that were materialized and the performance of the PCs
>
> In the original experiments, we select patch size 4 for the MAE model. That is, every 4*4 patch is materialized as a LV $Z$. To study the influence caused by the number of materialized LVs, we perform LVD using two new MAE models with patch size 2 and 8, respectively. The results are shown in the table below.
>
> ```text
> +================+======+=====+======+
> + patch size     +  2   +  4  +  8   +
> + ImageNet32 bpd + 4.57 + 4.38+ 4.44 +
> +================+======+=====+======+
> ```
>
> We observe that with patch size 8, the performance becomes slightly worse and with patch size 2 the performance becomes much worse. However, we highlight that the inferior performance of the patch-size-2 model is not because the use of more materialized latent variables. The reason is that the sub-PCs $p(\mathbf{x}_i \vert Z_i = j)$ suffer from worse likelihoods since the receptive field is limited to 2*2. In addition, the 2*2 patch MAE is harder to train and has larger RMSE compared to its 4*4 variant. Therefore, the performance of LVD is also determined by the deep generative model and the strategy to choose LVs.
>
> [1] Shih, Andy, Dorsa Sadigh, and Stefano Ermon. "HyperSPNs: compact and expressive probabilistic circuits." Advances in Neural Information Processing Systems 34 (2021): 8571-8582.
>
> [2] Peharz, Robert, et al. "Random sum-product networks: A simple and effective approach to probabilistic deep learning." Uncertainty in Artificial Intelligence. PMLR, 2020.

---

### Decision · Program_Chairs · 2023-01-20

**Decision:**

Accept: notable-top-5%

**Justification For Why Not Higher Score:**

N/A

**Justification For Why Not Lower Score:**

The reviewer scores are consistently high throughout all reviewing stages, with also extensive additional input from the authors to address thoroughly the reviewer's issues.

**Metareview: Summary, Strengths And Weaknesses:**

The paper describes an approach (using latent variables) to over come (most likely) optimisation issues in large-scale probabilistic circuits.  The work seems novel and the empirical results are compelling. One downside was a limited understanding of what factors make the method successful. This was addressed quite in-depth by the authors during the discussion phase and the reviewers were satisfied with the conclusions reached by the authors.

This is a well written high quality paper with a nice methodological contribution supported by extensive experiments.

**Note From Pc:**

if the above contains the word "oral" or "spotlight" please see: "oral" presentation means -> notable-top-5% and "spotlight" means -> notable-top-25%. As stated in our emails, we are disassociating presentation type from AC recommendations